# Care-seeking behaviours of mothers and associated factors for possible serious bacterial infection in young infants during COVID-19 pandemic in Ethiopia: mixed-methods formative research

Gizachew Tadele Tiruneh ![ORCID] ,[1,2] Lisa R Hirschhorn ![ORCID] ,[3] Nebreed Fesseha,[1] Dessalew Emaway,[1] Kristin Eifler,[4] Wuleta Betemariam[5]

For numbered affiliations see end of article.

**Correspondence to**
Dr Gizachew Tadele Tiruneh; gizachew_tadele@et.jsi.com

## ABSTRACT

**Objectives** Implementation research was employed to examine rates and contextual factors associated with mothers' care-seeking for their sick neonates and identify challenges for community-based possible serious bacterial infection (PSBI) services access and implementation during the COVID-19 pandemic.

**Design** We conducted formative research involving household survey and programmatic qualitative study.

**Setting** This formative study was conducted in Dembecha and Lume woredas of Amhara and Oromia regions.

**Participants** Data were captured from 4262 mothers aged 15–49 years who gave live birth 2–14 months before data collection, and interviews with 18 programme managers and 16 service providers in April to May 2021.

**Analysis** A multilevel regression model was employed to identify predictors of maternal care-seeking for PSBI and thematic qualitative analysis to inform strategy development to strengthen PSBI implementation.

**Results** Overall, 12% (95% CI 11.0% to 12.9%) and 8% (95% CI 7.9% to 9.6%) of mothers reported any newborn illness and severe neonatal infection (PSBI), respectively. More than half of mothers sought formal medical care, 56% (95% CI 50.7% to 60.8%) for PSBI. Women who received postnatal care within 6 weeks (adjusted OR (AOR) 2.08; 95% CI 1.12 to 3.87) and complete antenatal care (ie, weight measured, blood pressure taken, urine and blood tested) (AOR 2.04; 95% CI 1.12 to 3.75) had higher odds of care-seeking for PSBI. Conversely, fear of COVID-19 (AOR 0.27; 95% CI 0.15 to 0.47) and residing more than 2 hours of walking distance from the health centre (AOR 0.39; 95% CI 0.16 to 0.93) were negatively associated with care-seeking for severe newborn infection. Multiple pre-existing health system bottlenecks were identified from interviews as barriers to PSBI service delivery and exacerbated by the COVID-19 pandemic.

**Conclusion** We found gaps in and factors associated with care-seeking behaviour of mothers for their sick young infants including fear of COVID-19 and pre-existing health system-level barriers. The findings of the study were used to design and implement strategies to mitigate COVID-19 impacts on management of PSBI.

### STRENGTHS AND LIMITATIONS OF THIS STUDY

⇒ This study employed mixed-methods research to explore the perspectives and experiences of mothers, programme managers, service providers and community volunteers for an in-depth understanding of the limiting factors for the uptake and delivery of community-based management of sick young infants during the COVID-19 pandemic in Ethiopia.

⇒ The applications of implementation science frameworks offer helped us to systematically explore implementation challenges and design appropriate strategies to mitigate the impact of COVID-19 on the delivery and uptake of community-based management of possible serious bacterial infection (PSBI).

⇒ All possible barriers affecting the community-based management of the PSBI programme may not be captured as mothers' views were not included in the qualitative study.

⇒ Cross-sectional study design may not capture on-going challenges and impacts due to the dynamic nature of COVID-19 pandemic.

## BACKGROUND

While Ethiopia has seen a substantial decline in childhood mortality over the last two decades, the change in neonatal mortality rate (NMR) lagged behind the decrease in postneonatal and child mortality. The NMR in 2019 was 33 per 1000 live births, contributing to approximately 56% of under-five child mortalities.[1]

Possible serious bacterial infection (PSBI) is a leading cause of mortality in young infants.[2] WHO recommends that young infants aged 0–59 days with clinical signs of infection should receive hospital-based treatment.[3] However, existing evidence shows that in resource-limited settings many young infants with signs of infection do not receive the recommended inpatient treatment due

to limited access to affordable care. Evidence from low and middle-income countries (LMICs) has demonstrated the effectiveness of a simplified regimen of treatment, including injectable and oral antibiotics. Based on these studies, WHO recommends the management of sepsis with a simplified regimen outside of the hospital setting when referral is not feasible.[3]

In 2012, Ethiopia introduced integrated community case management of childhood illnesses and newborn care in its flagship Health Extension Program (HEP) to improve access to and use of appropriate treatment including for neonatal sepsis in young infants aged 0–59 days when referral is not possible.[4] Despite national efforts, uptake of sick newborn care is still very low. Between November 2020 and April 2022, the Bill & Melinda Gates Foundation-funded JSI Research & Training Institute/The Last Ten Kilometers Project (L10K) conducted implementation research in Ethiopia to understand the demand-side and supply-side challenges of management of PSBI when referral is not feasible during the COVID-19 pandemic. The project conducted formative work to understand challenges and inform implementation strategies, and provided technical support and monitoring to inform adaptation and progress. We describe results from the formative work on mothers' care-seeking behaviour and practices for their sick neonates during the pandemic, along with their determinants, and use them to inform the development of PSBI implementation strategies during and after the pandemic.

## METHODS
### Study design and data
This formative study was conducted in Dembecha and Lume woredas of Amhara and Oromia regions, respectively. The two woredas comprised 11 health centres, 66 health posts (HP) and an estimated 250 000 people. Details of the study settings and the health system can be found elsewhere.[5]

The study is a multiple methods research using the Implementation Research Logic Model (IRLM), a combination of Reach, Effectiveness, Adoption, Implementation and Maintenance (RE-AIM) and Consolidated Framework for Implementation Research (CFIR)[6] to understand the complex challenges to PSBI implementation before and during the pandemic; how implementation strategies need strengthening or adaptation and to influence quality delivery of PSBI and improve implementation outcomes; and how implementation strategies were designed, reported and synthesised to determine outcomes and learnings.

Data were collected through interview of mothers, programme managers, service providers and community volunteers from 18 April to 24 May 2021. The quantitative component of the study was a cross-sectional population-based study interviewing women who gave live birth 2–14 months before data collection whether their infants were sick during their infancy ages of 0–59 days or not. A structured questionnaire adapted from previous studies was used to capture the data. The questions were translated into local languages (Amharic and Oromiffa). The quantitative component also included HP assessments and self-administered interviews with health extension workers (HEWs), frontline health workers, available on the day of the visit. The HP assessment was conducted using an adapted tool from WHO Service Availability and Readiness Assessment.

The survey was used to collect information about household and sociodemographic characteristics of mothers, awareness of PSBI, history of any infection of their infants during the neonatal period (0–59 days), care-seeking for any infection of the infant, knowledge and risk perception of COVID-19, fear of COVID-19 infection to seek care and experiences related to the use of maternal and newborn health services. Prior to data collection, interviewers, supervisors and coordinators were thoroughly trained during a 1-day training on the questionnaire. A pretest of research instruments was also conducted. Survey supervisors and coordinators monitored and supervised the fieldwork and ensured data quality by randomly revisiting selected households to validate responses.

The qualitative component employed a programmatic qualitative study using the CFIR as included in the IRLM.[7] The framework was used to understand factors that influenced implementation outcomes, including inner (characteristics of the health system), intervention, actor (provider and caregivers) and outer settings. We used a semistructured questionnaire to conduct in-depth interviews (IDI) with health facility workers, HEWs and programme managers at each level of the health system, as well as with caretakers and community volunteers in the intervention kebeles. Interviews were conducted in Amharic and Oromiffa by experienced qualitative research consultants at a convenient place in the community, such as a community hall or the respondent's house.

### Patient and public involvement
Patients and communities were not involved in this study.

### Sample size and sampling technique
Our study contains the following assumptions: of 3% estimated annual live birth rate, 5% prevalence of newborn sepsis and 56% of newborn sepsis cases that sought medical care,[8] an estimated 375 neonatal sepsis cases and 210 cases that sought medical care could be obtained in the study woredas (out of an estimated population of 250 000). To calculate the sample size we assumed a 95% confidence level ($Z\alpha/2=1.96$), power of 80% and design effect of 1.5, given that 56% of mothers sought medical care for ill babies in the neonatal period.[8] Accordingly, all 66 kebeles were included in the study and relying on a census, we identified and interviewed mothers with infants aged 2–14 months. The household survey sample size was deemed adequate as it was estimated to double population formula for a comparative before-and-after cross-sectional study design with sufficient power to detect

a 20% change in sick newborn care-seeking behaviour due to the intervention.

We recruited all eligible households with children under 15 months of age from updated lists obtained from the Community Health Information System of all 66 HPs, which included information on households with infants under 15 months of age and their unique household identifiers. These records were available at HPs and stored in numerical order. When a complete list of households was not available at HPs, an ad hoc list of households was established by data collectors with the help of the HEWs, Women Development Armies (WDAs) and community volunteers before data collection.

For the qualitative component of the study, we conducted 34 IDIs including programme managers (6), development partners (4), service providers (16) and WDA volunteers (8).

## Measurement

The dependent variable of interest was the proportion of women who sought medical care from appropriate providers for their sick young infants during the first 2 months of life. Sick young infants were defined as those whose mothers reported symptoms of severe infection, including difficulty breathing, chest indrawing, no longer feeding well, being unusually hot or cold, being less active than usual and/or having convulsions during the first 2 months of life. The independent sociodemographic and obstetric characteristic variables included: maternal education, household wealth index, number of children, distance of residence from a health facility, utilisation of antenatal care (ANC) services, whether complete ANC was received, place of delivery, administrative residence (woreda) and level of fear of COVID-19 and its impact on seeking care.

ANC services were considered complete if the mother's blood pressure (BP) was taken, weight measured and blood and urine tested during the pregnancy of their gest infant. A wealth index score was constructed for each household with the principal component analysis of the household possessions (electricity, watch, radio, television, mobile phone, telephone, refrigerator, table, chair, bed, electric stove and kerosene lamp) and household characteristics (type of latrine, water source, floor and wall materials). Subsequently, households were ranked according to wealth score and then divided into five quintiles using the principal component analysis.[9] Mothers' fear of COVID-19 was measured by taking the maximum value of yes/no responses to each of the following questions: fear of infection during ANC or delivery visit, fear of fetus/baby being infected during ANC or delivery visit or fear of spreading infection. This variable was measured for those who delivered after the first case of COVID-19 was reported in Ethiopia in March 2020.

## Data analysis

Random intercept logistic models were fitted to estimate associations between the individual and community variables, and the likelihood of seeking care for PSBI. This enabled the probability of PSBI care-seeking to vary randomly across communities based on the assumption that the effects of individual characteristics were the same in each community (ie, the coefficients of all explanatory variables were fixed across communities using *xtlogit* and *xtmelogit* Stata commands). The null model, model 1, was fitted without the explanatory variables. In model 2, the intercept model was fitted with level 1 and level 2 variables. In model 3, a cross-level interaction between woreda of residence and distance to the health centres was fitted to reveal any evidence of effect modification of the association between distance to the health centre and PSBI care-seeking by woreda of residence. Explanatory variables were identified based on the CFIR and RE-AIM frameworks included in the models. We present the adjusted ORs (AOR) and CIs at the 95% level.

Qualitative interviews were audio taped and transcribed verbatim, and the transcript texts were exported to ATLAS.ti software for thematic analysis. Both deductive and inductive coding approaches were applied starting with a codebook reflecting the components of the IRLM (determinants, strategies, outcomes). Data were iteratively read and subsequently classified into one of the codes.

## RESULTS

A total of 4262 mothers of children aged 2–14 months were interviewed from the household survey. More than two-thirds of the respondents were 20–34 years old. The majority did not have any education (56%) and most lived 2 hours or more of walking distance from a health centre (table 1).

### Care-seeking for neonatal illness

In the household survey, we identified 508 sick young infants and 371 severe neonatal infection cases. Twelve per cent (11.6%) (95% CI 9.8% to 13.7%) of mothers reported a history of illness in their youngest child during their neonatal period with 8.4% (95% CI 7.0% to 10.0%) reporting symptoms of severe infection. A little more than half of mothers sought formal medical care, 57.6% (95% CI 47.2% to 67.4%) for any illness and 56.4% (95% CI 45.5% to 66.7%) for severe infection, primarily from health centres (figure 1).

### Barriers to delivery of PSBI services: COVID-19 and other pre-existing conditions

PSBI service registers from the 66 HPs showed a decline in sick young infant HP visits from 1675 cases in 2019 to 538 in 2021. Respondents highlighted several factors that caused low uptake and delivery of PSBI services, including (1) the COVID-19 pandemic, (2) low confidence and competence of HEWs, (3) unavailability of essential drugs, (4) weak support and linkage system and (5) suboptimal functionality of the WDA.

**Table 1** Background characteristics of the household survey participants, 2021

| Background characteristics | Frequency | % |
|---|---|---|
| Age of the mother | | |
| 15–19 | 211 | 5.0 |
| 20–34 | 2943 | 69.0 |
| 35–49 | 1108 | 26.0 |
| Maternal education | | |
| Cannot read/no school | 2365 | 55.5 |
| Primary | 853 | 20.0 |
| Secondary and higher | 1044 | 24.5 |
| Wealth quintile | | |
| Poorest | 805 | 18.9 |
| Poorer | 827 | 19.4 |
| Poor | 849 | 19.9 |
| Less poor | 879 | 20.6 |
| Least poor | 903 | 21.2 |
| Number of children | | |
| 1 | 1043 | 24.5 |
| 2–3 | 1565 | 36.7 |
| 4+ | 1653 | 38.8 |
| Distance to health centre, hours (walking distance) | | |
| <2 | 534 | 12.5 |
| ≥2 | 3728 | 87.5 |
| Woreda | | |
| Lume | 1768 | 41.5 |
| Dembecha | 2494 | 58.5 |
| Total number of respondents | 4262 | 100.0 |

### Influence of COVID-19 on maternal, newborn and child healthcare-seeking and delivery

The qualitative analysis found that the COVID-19 pandemic had a detrimental impact on care-seeking and delivery of PSBI services. We found that programme managers and service providers perceived that the pandemic generally affected all child health services, including PSBI activities related to generating awareness,

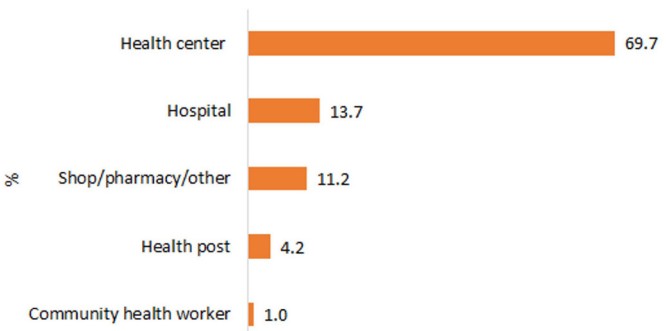

**Figure 1** Place of care-seeking for the possible serious bacterial infection (PSBI), 2021.

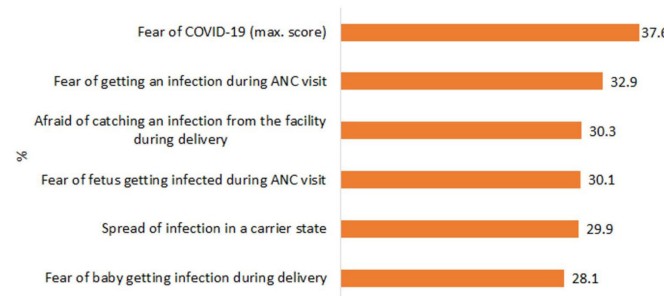

**Figure 2** COVID-19 factors impacting care-seeking, 2021. ANC, antenatal care.

or social and behavioural change communication (SBCC) efforts such as interrupted social gatherings, community meetings and community sensitisation programmes, particularly in the first 2–3 months of the pandemic in Ethiopia.

> The pandemic has affected all activities related to awareness creation or SBCC efforts. It affected advocacy works, community sensitization, pregnant women conferences, and other activities in that it made things very stagnant or not moving forward. (IDI-PM-MoH02)

One-third of mothers from the household survey reported fear of getting an infection in seeking maternal and newborn care (figure 2).

### Low confidence and competence of HEWs

Survey was completed in 79 of 117 (68%) HEWs that were available at the time of data collection. We found that only 41% knew all signs of PSBI. This finding was confirmed by programme managers and service providers who also perceived that low competence and confidence levels of HEWs to administer curative services were barriers to quality PSBI services.

> The Health Extension Workers' confidence is low… Since they had not practiced the curative services after they received the PSBI training, they are afraid of providing services, they are rather referring cases and not promoting the PSBI service. (IDI-PM-P02)

### Erratic supply of drugs and commodities

Similar to the facility surveys, informants affirmed that stock-out of drugs was a challenge to providing PSBI treatment services. Understocked HPs could not adequately meet PSBI service delivery requirements due to inaccurate forecasting and planning, insufficient medical supplies, inconsistent supply of drugs, drug expiration, failure to request and refill drug stocks and supplies and stock-outs.

Moreover, some essential drugs were not available in the local market, which meant that even when there was no stock in the facility, it was also not possible to procure drugs from local private sector sources. This also

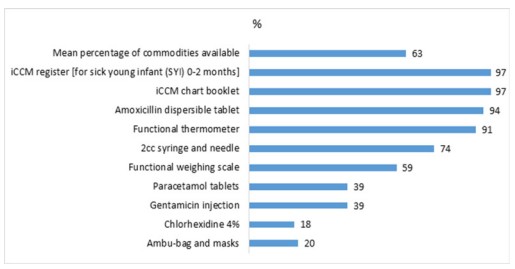

**Figure 3** Availability of essential drugs at surveyed health posts (HPs), 2021. iCCM, integrated community case management of childhood illnesses and newborn care.

led to insufficient or inconsistent medical supplies and stock-outs.

> Some essential drugs are not available in the local market. For example, amoxicillin dispersible syrups and gentamicin injection 20mg/2ml preparation are not available. (IDI-PM-P02)

The HP assessment also demonstrated that on the day of the visit, a significant number of HPs were inadequately stocked with essential commodities. Most lacked gentamicin injection, chlorhexidine gel and ambu bag and mask (figure 3).

### Weak support, integration and facility linkages

Declining mentorship or supervision, lack of training and motivation of HEWs and suboptimal ownership at each level were other factors attributed to weak PSBI programme implementation. Some HEWs had not had any PSBI training, suggesting the need for routine training for new HEWs, as well as refresher training for established workers.

> There are many HEWs who did not take training and the existing HEWs like me have forgotten the training, so training is needed to enhance the newborn treatment in the health post. In addition to this regular input, supply is needed. (IDI-SP-HEW03)

Programme ownership and sustained support from the government were also found to affect PSBI services. Programme support was strong and there was close follow-up when development partners were actively supporting the implementation of PSBI services. When PSBI became integrated with the national service package, PSBI service delivery was slowed due to overall system performance issues, including a lack of resources and competition with other programmes.

> It (PSBI) is not owned by regions, zones, or woredas. A little bit is better at the ministry level. It is not equally getting focus like EPI and other child health programs. Once the development partners' support phased out, mentorship and supervision activities are quit. (IDI-PM-P01)

The fragmented referral system was one of the challenges that led to the low uptake of PSBI services. In the referral system, weak linkages between health facilities resulted in insufficient communication, feedback or follow-up on referred cases. '…*the linkages between health post and health center became weaker and to the extent, HEWs are not provided referral slip as before*' (IDI-PM-MoH02).

### Suboptimal community engagement and low functionality of WDAs

Even when PSBI services were available at HPs, inadequate community-level service emerged as an important factor for the low demand and uptake of PSBI services.

> Community members may not be aware of the availability of newborn care at health post level because HEWs haven't fully disseminated information though they were directed to do so during PSBI trainings. (IDI-PM-MoH03)

WDA volunteers carried out community education and mobilisation activities and served as the HEW voice in communities. As a part of the community, they were engaged in early pregnancy identification, birth notification, health promotion, identifying and referring sick newborns and other social support activities. However, various factors contributed to a decline in WDA functionality. These included poor support and supervision, women's low literacy and negotiation skills, the perception of low commitment and that many had overlapping tasks to manage. '*The role of WDA is declining over time due to the decline of the program support. ….*' (IDI-PM-MoH01).

Ideally, all households would be included in the WDA networks across all kebeles. This, however, made the structure very broad and hard to manage. '*It is difficult to manage WDA as it is huge in number…*' (IDI-PM-P01).

### Predictors of care-seeking for PSBI

In the null model, the intraclass correlation coefficient showed that 23% of the residual variation in the propensity to uptake PSBI is attributable to unobserved community characteristics. The likelihood ratio test showed that there was strong evidence that the between-community variance was non-zero ($\chi^2$=41.25; p<0.01). Model 2, the intercept model, showed that women who received postnatal care (PNC) within 6 weeks and received complete ANC (ie, weight measured, BP taken, urine and blood tested) had higher odds of care-seeking for PSBI. Women who resided in Lume woreda had higher odds of seeking care for PSBI than their counterparts in Dembecha (AOR 2.79; 95% CI 1.17 to 6.66). Furthermore, fear of COVID-19 infection was negatively associated with uptake of care-seeking for PSBI (AOR 0.30; 95% CI 0.16 to 0.55). However, maternal education, household wealth and knowledge of COVID-19 were not significantly associated with mothers' care-seeking for PSBI.

Model 3, the cross-level interaction model, revealed evidence of effect modification of the association between distance to the health centre (>2 hours of walking distance) and PSBI uptake by woreda of residence. The within-community effect of the administrative woreda in Lume disappeared, while the within-community effect of distance to the health centre on the odds of receiving

PSBI was significantly lower for women who resided within 2 hours of walking distance to the health centre as compared with women who resided further away from the health centre (AOR 0.39; 95% CI 0.16 to 0.93) (table 2).

## DISCUSSION

This study revealed a high prevalence of severe neonatal infection and low care-seeking behaviour of mothers for their sick young infants in rural communities in Lume and Dembecha woredas of Ethiopia. Fear of COVID-19 infection negatively affected care-seeking for sick young infants. Out of 4262 participants, one-third of the mothers reported fear of getting COVID-19 infection in seeking care. Conversely, receiving ANC and PNC consultations and residing more than 2 hours of walking distance to the health centre predicted care-seeking for PSBI. In addition, multiple pre-existing health system factors were identified as barriers to PSBI service delivery.

Our results on prevalence are similar to other community-based studies in Ethiopia which identified a prevalence of PSBI ranging from 5% to 14%.[8 10–13] However, despite the availability of evidence-based, cost-effective interventions to manage neonatal infections,[14] only a little more than half of the Ethiopian mothers sought medical care for ill babies in the neonatal period.[8 13]

This study shows that fear of COVID-19 infection negatively affected care-seeking for their sick young infants in rural settings in Ethiopia. Previous modelling studies[15 16] and observational studies[17–20] showed that fear of contracting COVID-19 infection was the main reason for access to newborn care services and delayed health-seeking for child health services in LMICs including Ethiopia.[21 22] Analysis of continuity of maternal and child health service delivery amid COVID-19 in eight sub-Saharan countries revealed significant service disruptions in outpatient care and child vaccinations.[20] Similarly, a cross-sectional study in Ghana and South Africa showed a significant decline in clinic attendance and hospital admissions of children during the pandemic,[23 24] indicating fear of COVID-19 might intensify delays and reduce access to newborn care. These results are similar to a rapid assessment conducted in 2020 by JSI Research & Training Institute through L10K that showed the emergence of COVID-19 affected the uptake and delivery of essential maternal and newborn health; exceptionally low coverage was observed in under-five pneumonia treatment and under-five outpatient attendance.[25] The qualitative component of this current study showed that the pandemic affected child health services, including PSBI, in the first 2–3 months of the pandemic. It showed that the pandemic continued to affect activities related to generating awareness or SBCC efforts, such as interrupting social gatherings, community meetings and community sensitisation programmes.[26]

Maternal healthcare uptake and receipt of quality ANC services were significantly associated with care-seeking for PSBI. Evidence from other studies has shown that uptake of the continuum of care services was associated with the uptake of newborn and child health services[27–29] and their survival.[30] Antepartum and postpartum care offered the opportunity for a woman to establish an informal forum to discuss the benefits of skilled care throughout their pregnancies and child-rearing periods.[31 32] Evidence from previous studies indicates that attending the recommended number of ANC and postpartum visits provided women with a chance to receive quality care, and more counselling during pregnancy and postpartum care regarding newborn care.[33 34] It also improved women's trust in providers and increased their positive experience with them, which was found to increase a woman's odds of receiving a continuum of maternal and child health services.[32]

Our cross-interaction model showed that woreda of residence modified the association between distance to health centre and uptake of PSBI care. There was variability in care-seeking for PSBI by residence, and shorter distance to the health centre was positively associated with mothers' care-seeking for their sick young infants. This suggests the existence of differential geographic access to PSBI, despite the programme being pro-poor, as documented in other studies regarding access to postpartum care in Ethiopia.[35 36] In addition, despite the expansion of primary care services in Ethiopia, distance remained a major barrier for many people. Previous studies reported that distance to reach health facilities, topography and lack of transportation deterred utilisation for treating sick young infants.[37–41] Evidence from developing countries also suggested uptake of services when they were more proximate.[42] These examples underline the need for expansion of health facilities and the design of appropriate strategies to reach disadvantaged communities and their newborns in Ethiopia.

This study demonstrated the unprecedented challenges posed by the COVID-19 pandemic on PSBI care-seeking during the pandemic in Ethiopia. The following pre-existing conditions proved to be the most formidable barriers and were also identified as challenges in previous studies: poor integration of PSBI in the woreda workstream when partners' support was phased out, weak support systems, erratic supply and poor logistics management systems, low competence and motivation of HEWs and low community demand.[10 38 39 43–47] We found that these challenges could be mitigated with several strategies, including integration of PSBI in the district health system workstream, integration of COVID-19 services with the primary healthcare services, continuous on-site coaching and training of HEWs, strong support system from health centres and woreda health offices, the use of participatory design and implementation methods, the strengthening of the supply chain system and the presence of community-level SBCC activities. These approaches were effective in engaging community volunteers, HEWs and the primary healthcare system and in better integrating PSBI for sustained implementation and scale-up.

**Table 2** Predictors of care-seeking for young infants with severe neonatal infection, multilevel logistic intercept models, 2021

| Predictors | Seek care for PSBI | Model 1: null model | Model 2: intercept model | | Model 3: cross-level interaction | |
|---|---|---|---|---|---|---|
| | | | AOR (95% CI) | P value | AOR (95% CI) | P value |
| **Maternal education** | | | | | | |
| Cannot read/no school | 109 (50.2) | | 1.00 | | 1.00 | |
| Primary | 41 (60.3) | | 1.47 (0.74 to 2.94) | 0.270 | 1.45 (0.73 to 2.91) | 0.294 |
| Secondary and higher | 57 (66.3) | | 1.48 (0.75 to 2.92) | 0.263 | 1.50 (0.76 to 2.98) | 0.247 |
| **Wealth quintile** | | | | | | |
| Poorest | 32 (41.0) | | 1.00 | | 1.00 | |
| Poorer | 45 (54.9) | | 1.40 (0.64 to 3.08) | 0.401 | 1.38 (0.63 to 3.04) | 0.424 |
| Poor | 48 (57.8) | | 1.62 (0.72 to 3.63) | 0.245 | 1.56 (0.69 to 3.53) | 0.285 |
| Less poor | 35 (61.4) | | 1.85 (0.73 to 4.72) | 0.198 | 1.77 (0.69 to 4.56) | 0.235 |
| Least poor | 47 (66.2) | | 1.34 (0.57 to 3.59) | 0.567 | 1.36 (0.50 to 3.69) | 0.549 |
| **Received ANC** | | | | | | |
| Yes | 174 (61.1) | | 1.76 (0.85 to 3.66) | 0.131 | 1.77 (0.85 to 3.71) | 0.129 |
| No | 33 (38.4) | | 1.00 | | 1.00 | |
| **Complete ANC** | | | | | | |
| Yes | 123 (67.2) | | 2.08 (1.14 to 3.79) | 0.018 | 2.04 (1.12 to 3.75) | 0.021 |
| No | 84 (44.7) | | 1.00 | | 1.00 | |
| **Place of delivery** | | | | | | |
| Home | 52 (41.6) | | 1.00 | | 1.00 | |
| Facility | 155 (63.0) | | 1.18 (0.63 to 2.19) | 0.603 | 1.21 (0.65 to 2.27) | 0.542 |
| **Any PNC within 6 weeks** | | | | | | |
| Yes | 80 (72.7) | | 2.15 (1.16 to 3.98) | 0.015 | 2.08 (1.12 to 3.87) | 0.021 |
| No | 127 (48.7) | | 1.00 | | 1.00 | |
| **Fear of COVID-19** | | | | | | |
| Yes | 69 (44.2) | | 0.30 (0.16 to 0.55) | <0.01 | 0.27 (0.15 to 0.47) | <0.01 |
| No | 138 (64.2) | | 1.00 | | 1.00 | |
| Distance to the health centre, hours (walking distance) | | | | | | |
| ≤2 | 47 (73.4) | | 1.00 | | 1.00 | |
| >2 | 160 (52.1) | | 0.51 (0.23 to 1.12) | 0.095 | 0.39 (0.16 to 0.93) | 0.033 |
| **Woreda of residence** | | | | | | |
| Dembecha | 156 (52.4) | | 1.00 | | 1.00 | |
| Lume | 51 (69.9) | | 2.79 (1.17 to 6.66) | 0.020 | 0.62 (0.09 to 4.23) | 0.624 |
| Distance to health facility* Woreda* | | | | | | |
| >2 hours* Lume | | | | | 5.69 (0.74 to 43.50) | 0.094 |
| Community-level intercepts (SE) | | 1.68 (0.35) | 0.69 (0.39) | | 0.89 (0.53) | |
| *Random effects* | | | | | | |
| Community-level variance (SE) | | 0.99 (0.40) | 0.58 (0.30) | | 0.60 (0.31) | |
| Log-likelihood ratio test | | 41.25 | 12.81 | | 12.67 | |
| *Model fit statistics* | | | | | | |
| ICC (SE) | | 0.23 (0.07) | 0.15 (0.07) | | 0.15 (0.07) | |
| AIC | | 472.07 | 442.58 | | 441.92 | |

Continued

**Table 2** Continued

| Predictors | Seek care for PSBI | Model 1: null model | Model 2: intercept model | | Model 3: cross-level interaction | |
|---|---|---|---|---|---|---|
| | | | AOR (95% CI) | P value | AOR (95% CI) | P value |

*indicates interaction between distance to health facility and woreda of residence
AIC, Akaike information criterion; ANC, antenatal care; AOR, adjusted OR; ICC, intraclass correlation coefficient; PNC, postnatal care; PSBI, possible serious bacterial infection.

Our study had limitations. Like any survey, responses were potentially influenced by recall and social desirability bias. Thus, to improve reporting, memory aids were used. Though we tried to triangulate data from programme managers, service providers and community volunteers, qualitative data were not collected from women which would limit the findings of this study. Another limitation would be the qualitative findings are subjective and confounded by the individual's prevailing contexts.

## CONCLUSIONS

Care-seeking for sick young infants is influenced by physical access, antepartum and postpartum care service delivery, health system bottlenecks and fear of COVID-19. Accordingly, there is a need to strengthen the primary health system to be more resilient to provide quality PSBI services during and after the COVID-19 pandemic. Evidence-informed SBCC interventions are pivotal to providing current and comprehensive information on transmission and prevention of COVID-19 in multiple and accessible formats to boost the confidence of clients and mitigate fear of COVID-19 infection. Active engagement of communities to create continuous community awareness activities, actively identify pregnancies, notify births to HEWs, identify sick newborns and link them to the HEP is critical to maintaining service delivery and reducing the negative impacts of COVID-19 on newborn health outcomes.

**Author affiliations**
[1]Improving Primary Healthcare Project, JSI Research & Training Institute, Addis Ababa, Ethiopia
[2]Behavioral Science, Addis Continental Institute of Public Health, Addis Ababa, Ethiopia
[3]Feinberg School of Medicine and Havey Institute of Global Health, Northwestern University, Chicago, Illinois, USA
[4]International Division, JSI Research & Training Institute, Boston, Massachusetts, USA
[5]Center for Healthy Women, Children, and Communities, JSI Research & Training Institute, Washington, DC, USA

**Acknowledgements** We thank the Bill & Melinda Gates Foundation (BMGF) for funding this implementation research. The implementation of this survey would not have been possible without the support of the Ministry of Health and Amhara and Oromia Regional Health Bureaus. We acknowledge the interviewers and the supervisors for their hard work, dedication and for accomplishing the fieldwork on schedule. Finally, we take this opportunity to extend our gratitude to all study participants who took their time to respond to the survey questionnaires and provide us with invaluable information. Alexandra Kamberos is acknowledged for editing the manuscript.

**Contributors** GTT, NF, DE, KE and WB conceived the study design and methods. GTT, NF, DE, KE, WB and LRH formulated the research question, analysis method, interpretation and critical review. GTT drafted the manuscript. WB received the grant. All authors read and approved the final manuscript. GTT, the corresponding author, is assigned as a garantor.

**Funding** The article write-up and publication fee were supported by the Bill & Melinda Gates Foundation (Investment No INV-024241). JSI Research & Training Institute has supported us in the form of salaries for authors (GTT, DE, NF, WB and KE).

**Competing interests** The authors (GTT, NF, DE, WB and KE) work for JSI Research & Training Institute, a commercial company.

**Patient and public involvement** Patients and/or the public were not involved in the design, or conduct, or reporting, or dissemination plans of this research.

**Patient consent for publication** Consent obtained from parent(s)/guardian(s).

**Ethics approval** This study involves human participants and was approved by the Ethiopian Public Health Association (EPHA) Research Ethics Review Board (Reference No EPHA/OG/166/21, dated 16 April 2021). Participants gave informed consent to participate in the study before taking part.

**Provenance and peer review** Not commissioned; externally peer reviewed.

**Data availability statement** Data are available upon reasonable request.

**ORCID iDs**
Gizachew Tadele Tiruneh http://orcid.org/0000-0002-5842-9518
Lisa R Hirschhorn http://orcid.org/0000-0002-4355-7437

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
