## [Reviewer comments · BMJ Open]

ARTICLE DETAILS

TITLE (PROVISIONAL)	Care-seeking behaviors of mothers and associated factors for possible serious bacterial infection in young infants during COVID-19 pandemic in Ethiopia: mixed-methods formative research
AUTHORS	Tiruneh, Gizachew; Hirschhorn, Lisa; Fesseha, Nebreed; Emaway, Dessalew; Eifler, Kristin; Betemariam, Wuleta

VERSION 1 – REVIEW

REVIEWER	Bedir Demirdağ, Tuğba Gazi University Faculty of Medicine, Pediatric Infectious Diseases
REVIEW RETURNED	27-Mar-2023

GENERAL COMMENTS	I congratulate the authors for this very valuable research, I am sure that the results of this study will help the scientific community about influences of physical access, antepartum and postpartum care service delivery, health system bottlenecks, and fear of COVID-19 on care-seeking of infants. Besides, the study emphasizes the deficiencies of primary health care services, especially in developing or underdeveloped countries. Accordingly, there is a need to strengthen the primary health system. I think the study is worth to publish in this journal.
---

REVIEWER	Neill, Sarah University of Plymouth, School of Nursing and Midwifery
REVIEW RETURNED	04-Jun-2023

GENERAL COMMENTS	Thank you for submitting this paper - the focus on the affect of the pandemic on health care seeking for a sick infant in a LMIC is an important one for all LMICs. However your paper seems to focus more on the health services used and provided than on the stated focus in the title. Abstract - Your abstract sets the scene but you describe this as implementation research but the remainder of the abstract sets it out as an observational mixed-methods study. This could also be described as a theory development stage in the development of a complex intervention (See UK Medical Research Council guidance). In the main text you explain that this project used the Implementation Research Logic Model - it would be good to make this clear in the abstract. There is no clear statement of the aim of the project whose findings you are reporting here. Methods - here you refer to 'The overall study' - this suggests that this paper is only reporting one part of a bigger study. Was this the case? If yes, say so. If not, rephrase so that it is clearer. Its not clear why the specific age group of infants was included - technically an infant is a baby aged from 1 month to 1 year of age - your chosen age range was 2-14 months which does not align.
--

	How was the questionnaire developed? Did it already exist? Or did you create it? If so what was the evidence base for choosing those questions? Results - The results are wide ranging beyond the initial stated focus of the paper. I wonder if you need to reconsider the title of the paper to make it clear this is not just about help-seeking during COVID-19. In addition the data as presented does not seem to show whether or not fear of COVID-19 affected help-seeking for a sick infant - just that mothers were fearful of the infection and that there seems to be correlation. This was a missed opportunity to ask each mother directly. Perhaps you were trying to put too much into one paper. The single sentence on limitations is insufficient - there are other limitations inherent in the survey (how was it developed?) and in the rather lengthy and highly structured interview schedule which should be considered. Choosing not to collect any qualitative data from mothers has also limited the findings of the research and should be acknowledged. I don't find the data files helpful as an inclusion here - it is your job as the research team to do the analysis - not the reader's. If you want to share the data then you need to publish it in a data repository, not with your paper. Appendices need headings and all need to be referred to in the text with that heading. It would be helpful if the questionnaire was included as a PDF rather than as an excel file as it would make it easier to see the structure of the questionnaire. There are also too many abbreviations used - these make the paper less easy to read.
--	--

VERSION 1 – AUTHOR RESPONSE

Reviewer: 1

Dr. Tuğba Bedir Demirdağ, Gazi University Faculty of Medicine

Comments to the Author:

I congratulate the authors for this very valuable research, I am sure that the results of this study will help the scientific community about influences of physical access, antepartum and postpartum care service delivery, health system bottlenecks, and fear of COVID-19 on care-seeking of infants. Besides, the study emphasizes the deficiencies of primary health care services, especially in developing or underdeveloped countries. Accordingly, there is a need to strengthen the primary health system.

I think the study is worth to publish in this journal.

Response: Many thanks your kind words and your time reviewing our paper.

Reviewer: 2

Prof. Sarah Neill, University of Plymouth

Comments to the Author:

Thank you for submitting this paper - the focus on the affect of the pandemic on health care seeking for a sick infant in a LMIC is an important one for all LMICs. However your paper seems to focus more on the health services used and provided than on the stated focus in the title.

Response: We agree and have edited it. The title is now edited as such "Care-seeking behaviors of mothers and associated factors for possible serious bacterial infection in young infants during COVID-19 pandemic in Ethiopia: mixed-methods formative research"

Abstract - Your abstract sets the scene but you describe this as implementation research but the remainder of the abstract sets it out as an observational mixed-methods study. This could also be

described as a theory development stage in the development of a complex intervention (See UK Medical Research Council guidance). In the main text you explain that this project used the Implementation Research Logic Model - it would be good to make this clear in the abstract. There is no clear statement of the aim of the project whose findings you are reporting here.

Response: This study was formative work for the implementation research project that was implemented for two years. The findings of the formative work were used to design and implement strategies to mitigate COVID-19's impact on the uptake and delivery of community-based management of newborn infections. This is now indicated in the abstract as suggested.

Methods - here you refer to 'The overall study' - this suggests that this paper is only reporting one part of a bigger study. Was this the case? If yes, say so. If not, rephrase so that it is clearer.

Response: Yes, this study used the formative assessment findings of the before-after evaluation design. We have clarified that this was only the formative assessment. Page 5, Lines 80-88

Its not clear why the specific age group of infants was included - technically an infant is a baby aged from 1 month to 1 year of age - your chosen age range was 2-14 months which does not align. How was the questionnaire developed? Did it already exist? Or did you create it? If so what was the evidence base for choosing those questions?

Response: The age group of infants for this study was under two months (0-59 days) age young infants who were the focus on the WHO and Ethiopian guidelines for PSBI diagnosis and treatment. We interviewed mothers who gave birth between 2-14 months, to limit to a 12-month recall period or annual birth cohort. Questions for mothers whether their infant was sick during the neonatal (2 months of age) period. Under-2-month age infants were excluded as it would underestimate the prevalence of infection as they could still develop PSBI. In this version, we clarified this to avoid confusions.

It now reads as "Data were collected from 18 April-24 May 2021. The quantitative component of the study was a cross-sectional population-based study interviewing women who gave live birth 2-14 months before data collection whether their infants were sick during their infancy age 0-59 days or not. A structured questionnaire adapted from previous studies was used to capture the data. The questions were translated into local languages (Amharic and Oromifa) [7, 8]." Page 5, lines 89-94

Results - The results are wide ranging beyond the initial stated focus of the paper. I wonder if you need to reconsider the title of the paper to make it clear this is not just about help-seeking during COVID-19. In additional the data as presented does not seem to show whether or not fear of COVID-19 affected help-seeking for a sick infant - just that mothers were fearful of the infection and that there seems to be correlation. This was a missed opportunity to ask each mother directly.

Response: The title of the paper is now edited to reflect the focus of the paper. Furthermore, we clarified in the data collection section that we directly interviewed mothers whether their fear of COVID-19 affected care-seeking for their sick young infant or not. Pages 5 and 6, Lines 98-105 Perhaps you were trying to put too much into one paper. The single sentence on limitations is insufficient - there are other limitations inherent in the survey (how was it developed?) and in the rather lengthy and highly structured interview schedule which should be considered. Choosing not to collect any qualitative data from mothers has also limited the findings of the research and should be acknowledged.

Response: Many thanks and we agree. We have expanded the limitations section to included additional challenges and limitations.

Lines 334-38 reads "Our study had limitations. Like any survey, responses were potentially influenced by recall and social desirability bias. Thus, to improve reporting, memory aids were used. Though we tried to triangulate data from program managers, service providers, and community volunteers, qualitative data were not collected from women which would limit the findings of this study. Another limitation would be the qualitative findings are subjective and confounded by the individual's prevailing contexts."

I don't find the data files helpful as an inclusion here - it is your job as the research team to do the analysis - not the reader's. If you want to share the data then you need to publish it in a data

repository, not with your paper. Appendices need headings and all need to be referred to in the text with that heading. It would be helpful if the questionnaire was included as a PDF rather than as an excel file as it would make it easier to see the structure of the questionnaire. There are also too many abbreviations used - these make the paper less easy to read.

Response: Many thanks. We have cited the recent published paper which published the questionnaire as supplementary material "Tiruneh GT, Nigatu TG, Magge H, Hirschhorn LR. Using the Implementation Research Logic Model to design and implement community-based management of possible serious bacterial infection during COVID-19 pandemic in Ethiopia. BMC Health Services Research. 2022;22(1):1515. doi: 10.1186/s12913-022-08945-9."